# Evaluation of Knowledge and Risk Perception about Antibiotic Resistance in Biology and Mathematics Young Students in Nîmes University in France

**DOI:** 10.3390/ijerph18189692

**Published:** 2021-09-14

**Authors:** Valentin Duvauchelle, Elsa Causse, Julien Michon, Patrick Rateau, Karine Weiss, Patrick Meffre, Zohra Benfodda

**Affiliations:** 1UPR CHROME, Université de Nîmes, CEDEX 1, F-30021 Nîmes, France; valentin.duvauchelle@unimes.fr (V.D.); elsa.causse@unimes.fr (E.C.); julien.michon@unimes.fr (J.M.); karine.weiss@unimes.fr (K.W.); patrick.meffre@unimes.fr (P.M.); 2Département de Psychologie, Université Paul-Valéry Montpellier 3, EPSYLON EA 4556, F-34000 Montpellier, France; patrick.rateau@univ-montp3.fr

**Keywords:** antibiotic, resistance, rational use, knowledge surveys, students, risk perception

## Abstract

In response to the antimicrobial resistance issue, the World Health Organization developed and conducted a survey in 2015 dealing with habits, antibiotic use, awareness of appropriate use and sensitization to the issue of antibacterial resistance. In France, we conducted a similar survey to investigate the use of antibiotics and students’ perceptions of the antibiotic resistance risk. Our results indicated that antibiotics are moderately taken (42% in the last six months), but mistakes remain in appropriate practices and knowledge. Many people still believe that the body develops resistance to antibiotics and 24% responded that antibiotics can be stopped before the end of the treatment if they feel better. Furthermore, only 14% said antibiotics could be used to treat gonorrhea while 57% indicated that influenza could be treated with antibiotics. We looked at risk perception as well, and noticed that students in biology were more aware of risk (mean score = 48.87) and health consequences (mean score = 40.33) than mathematics students (mean score = 44.11 and 37.44). They were more aware of the threat, had a better understanding of antibiotic resistance and their denial of this risk was less significant (mean score = 27.04 against 23.81). However, the importance of providing a minimum level of knowledge to young students has been emphasized, regardless of the field of expertise.

## 1. Introduction

Antibiotic-resistant organisms have emerged as a major public health concern, particularly in healthcare centers [1]. Bacteria have innate resistance mechanisms, but they can also develop resistance mechanisms to resist antibiotics [2,3]. Antibiotics have saved many lives, yet antibiotic resistance (ABR) has become a global issue [4]. ABR is ancient, natural and existed before the discovery of modern antibiotics in the 1940s [5]. However, antibiotic overuse creates selection pressure, forcing infection prevention and control strategies to prevent the spread of multidrug resistant infections [6,7]. ABR genes (ARGs), which are found in soil and water, are the most common source of ABR in the environment [8]. Institutions have been informed of the number of potential deaths and economic costs that this public health issue could cause. [9,10,11,12]. Hence, the World Health Organization (WHO) described a lack of new innovative antibiotics in 2019 and emphasized the need to change the way antibiotics are currently administered, as well as the difficulties of medical procedures such as caesarean sections and hip replacements [10,13]. This emerging risk, with a slow or non-existent reversibility rate [14], can be addressed through appropriate antibiotic use (i.e., dosage adherence, treatment completion, and the absence of self-medication), control strategies and education of the population on appropriate practices to reduce the spread of resistance [15].

According to previous research, less than half of Europeans are aware that antibiotics are ineffective against viruses [16]. Results of surveys of doctors in the Chicago area showed that only 60% of doctors supported restricting the use of broad-spectrum antibiotics [17]. In addition, work on junior doctors has been done previously in France and Scotland [18]. Their findings revealed that only 26% of this population are aware of the true prevalence of antibiotic misuse.

Beyond work on professionals, the WHO has decided to initiate work on risk perception and uses of the population in selected countries in 2015 [19]. The WHO survey was conducted with 9772 respondents from 12 countries, with two countries per continent. These countries were included in their study: Nigeria and South Africa for the African region, Barbados and Mexico for the American region, India and Indonesia for the Southeast Asian region, Russian Federation and Serbia for the European region, Egypt and Sudan for the Eastern Mediterranean region and China and Vietnam for the Western Pacific region. Their purpose was to assess knowledge about the appropriate use of antibiotics worldwide, with a focus on awareness of ABR risk.

As the authors explained, the limitations of their survey lie in the fact that only two countries were picked per “region”. In Europe, Serbia and Russia were selected. The lack of participants from West European countries has been partially overcome by Prigitano and co-workers in 2018 [20]. They did similar work in Italy with junior and senior students. By modifying the original WHO questionnaire entitled “Antibiotic resistance: multicountry public awareness survey”, they chose to also investigate personal use of antibiotics, knowledge of antibiotics and perception of the phenomenon of resistance. A total of 797 students in the medical field in Milan answered this survey.

France was not included in these former works. France remains a country with a high antibiotic consumption rate in Europe, despite recent campaigns to reduce inappropriate prescription [21,22]. It is therefore necessary to look at the practices, knowledge and perception of this risk in the population in order to better understand this phenomenon. Nevertheless, studies about ABR need to be continued in order to expand knowledge of ABR perception. In this article, we have performed similar work based on the WHO survey and the previous Italian study [20]. The first purpose was to study another country in Western Europe to improve WHO results. Furthermore, improvements have been made in the questionnaire to allow finer interpretations using Likert scales instead of true/false answers. In fact, the former works were only descriptive [19,20]. However, ABR is an emerging risk faced with a lack of knowledge on the part of the population, but also with a pre-established social perception. This is why we worked on knowledge on the one hand, and on social perception and self-reported behaviors on the other. This work was enhanced by the addition of these two complementary perspectives, which added a new dimension to previous work that focused on the description of uses and knowledge.

Three hypotheses were formulated. The first one was that subjects with inappropriate practices (i.e., non-adherence to dosage and self-medication) would have a lower perception of ABR risk. They would express less concern about health consequences of this risk and they would express more denial about this risk than subjects with appropriate practices (i.e., dosage adherence, treatment completion, and the absence of self-medication). The second hypothesis was that the more the risk would be perceived, the less people would express denial and the more they would express concern about health consequences of ABR. The third hypothesis was that subjects with a higher expertise in life sciences especially in microbiology and virology would have a higher perception of ABR risk and more concern about health consequences of this risk while having a lower denial of the phenomenon of ABR risk.

## 2. Materials and Methods

### 2.1. Participants

The survey was carried out on 310 young students in science (first- to third-year students, 114 women (36.8%) and 196 men (63.2%); mean age = 19.79, SD = 1.74, range 18–34). Among these students, 254 (81.9%) were in life sciences and 56 (18.1%) were in mathematics. A total of 79 women (31.1%) and 175 men (68.9%) were in life sciences; mean age = 19.86, SD = 1.83, range 18–34. A total of 21 women (37.5%) and 35 men (62.5%) were in mathematics; mean age = 19.46, SD = 1.25, range 18–25.

Life sciences students can be considered as “experts” compared to mathematics students, who can be considered as “naive” in our study. All of the 310 respondents were students at the University of Nîmes, south France in an urban area. All the students present during our class interventions answered our questionnaire and none refused.

### 2.2. Measures

G*Power software was used for each statistical analysis in the study. We used the same protocol as Prigitano and colleagues [20], based on the original WHO questionnaire while including some modifications to the measures they proposed and adding a supplementary scale (see Appendix A, Appendix A). The survey was divided on 3 parts: the first part dealing with uses and general knowledge was constructed of 7 closed questions (questions #1 to #7). In a second part, participants were asked to answer one scale in a counterbalanced order composed of 20 items (question #8). These items are based on the 19 items proposed by Prigitano and colleagues (and one additional item we included in this scale), but instead of true/false answers, participants answered on 6-point Likert scales (from 1 = “not agree at all” to 6 = “completely agree”). This scale included ten negative items (items 2, 6, 7, 8, 10, 14, 15, 17, 18 and 20) which were reverse-coded prior to analysis (see Appendix A, Appendix A for raw data and explanatory information).

The first scale (question #8) initially presented satisfactory internal reliability (α = 0.69).

The third part was added to the protocol and measured general ABR risk perception and was composed of 8 items (question #9) with 10-point Likert scales (from 1 = “not agree at all” to 10 = “completely agree”). This scale has been added based on previous works [23,24].

Bartlett’s Test of Sphericity (χ^2^(190) = 646.48, *p* < 0.001) and the KMO index (0.743) both supported Exploratory Factor Analyses (EFA) conducted on the first scale. The initial unrotated solution obtained with Principal Component Analysis (PCA) revealed satisfactory communalities (>0.300). The first factor of this initial unrotated solution explained 17.05% of the total variance, which led us to pursue analyses to explore the potential multidimensional structure of the scale. 

Next, a rotated solution was calculated (with Varimax rotation) which converged in 6 iterations. However, the scree test suggested a possible factor structure composed of two main dimensions. We then computed a rotated solution based on two dimensions. In this rotated solution, items 10, 16 and 17 loaded <0.300 and were suppressed. In the next rotated solution items 3 and 15 loaded <0.300 and were suppressed.

In the final rotated solution, items 1, 4, 5, 6, 9, 11, 12 and 13 loaded >0.300 on the first factor, which explained 17.28% of the total variance after rotation and items 2, 7, 8, 14, 18 and 19 loaded >0.300 on the second factor, which explained 13.66% of the total variance after rotation (Table 1). Item 20 was suppressed because of its poor contribution in terms of meaning.

The first factor presented acceptable internal reliability (α = 0.64) but the second factor presented a weaker internal reliability (α = 0.56).

These exploratory analyses allowed us to identify a first dimension dealing with the concern about the consequences of ABR risk in the health domain and a second dimension dealing with the denial of ABR risk. The dimension of concern for health consequences was composed of 8 items such as “if bacteria are resistant to antibiotics, it can be very difficult or impossible to treat the infections they cause” or “many infections are becoming increasingly resistant to treatment by antibiotics”. The denial of ABR dimension was composed of 6 items such as “I am not concerned about the impact of bacterial resistance on my health or that of my family” or “antibiotic resistance is only a problem for people who take antibiotics regularly”.

Concerning the dimension of concern for health consequences of ABR risk, the higher the score is, the higher the concern for health consequences of ABR risk is. For the dimension of denial of ABR risk, the higher the score is, the lower the denial of the ABR risk is.

Bartlett’s Test of Sphericity (χ^2^(28) = 349.97, *p* < 0.001) and the KMO index (0.658) both supported Exploratory Factor Analyses (EFA) conducted on the second scale. 

The initial unrotated solution obtained with Principal Component Analysis (PCA) revealed satisfactory communalities (>0.300) for all items (except item 6) which contributed to the first factor of this initial unrotated solution which explained 29.63% of the total variance, which led us to consider this scale as unidimensional (Table 2). We computed another unrotated solution without item 6. In this final solution, all items contributed >0.300 to a single factor, which explained 33.63% of the total variance. This scale presented an acceptable internal reliability (α = 0.64) and included items such as “I consider that antibiotic resistance represents a risk”. The higher the score is on the risk perception scale, the higher the risk perception is.

## 3. Results

### 3.1. Uses and Knowledge

A total of 129 people (42%) took antibiotics in the last six months, with 42 people (14%) in the past month (Figure 1, #1). These results are consistent with those described previously [19,20]. A total of 84% of respondents were prescribed antibiotics by a doctor and 85% received advice from a health professional (Figure 1, #2, #3). A total of 24% still think you can stop antibiotic treatment when you feel better (Figure 1, #4). This proportion was much higher than Prigitano’s results (4%), but in agreement with the WHO’s (25%). A slight majority (67%) considered you cannot take former antibiotics if it is to treat the same illness compared to 90% of Prigitano’s and 49% of WHO’s (Figure 1, #5). Answers for question #6 are in agreement with the WHO’s study (Figure 1, #6).

For question #7, there was a list containing 13 pathologies or symptoms. Four of them were treatable with antibiotics: gonorrhea, bladder or urinary infection, skin infection and traumatic wound (circled in green, Figure 2). Among the nine others, some diseases were caused by viruses (HIV, flu, measles), parasites (malaria), or were just symptoms (diarrhea, fever, sore throat, body aches, headache). A total of 10% of our respondents gave only good answers but did not quote them all. Only one respondent gave all the good answers. A total of 73% gave at least one good answer mixed with wrong ones and 16% gave only wrong answers. Only 14% considered gonorrhea could be treated with antibiotics. A large majority selected bladder infection is treatable with antibiotics (75%). Some symptoms were thought to be treated with antibiotics for respondents, like diarrhea (29%), fever (38%) and sore throat (38%). As well, some diseases caused by parasites or viruses were selected by students, like HIV (6%), flu (57%), malaria (17%) and measles (27%).

### 3.2. Risk Perception and Health Consequences Concern

We worked on risk perception measures. Figure 3 shows some trends based on average scores on selected items. Students appeared to be aware that there is a risk in France (Figure 3, items 1, 4, 6 and 9) and agree with appropriate practices when they are proposed (Figure 3, items 3 and 12). 

However, a lack of knowledge is highlighted with expertise questions about ABR and its transmission, as shown in Figure 4.

In line with our first hypothesis (Table 3), participants with the correct answer to question #4 (After starting treatment, when do you think you should stop taking antibiotics?) expressed less denial about ABR risk (i.e., they had a higher score on the denial scale than participants with the incorrect answer. They also showed more concern for health consequences of ABR risk.

Participants with the correct answer for question #5 (Do you think that antibiotics that have been given to others (a friend or family member) can be used as long as they have been used to treat the same illness?) expressed more concern for health consequences of ABR risk than participants with the incorrect answer and had a higher perception of ABR risk than participants with the incorrect answers.

Participants with the correct answer for question #6 (Do you think you can ask a doctor for the same antibiotics, if they have treated the same symptoms in the past?) expressed more concern for health consequences of ABR risk than participants with the incorrect answer did. They expressed less denial about ABR risk than participants with the incorrect answer and a higher perception of ABR risk than participants with the incorrect answers did (data summed up in Table 3).

Consistent with our second hypothesis (the more the risk is perceived, the less denial is expressed and the more concern about it there is), results revealed a moderate and positive relation between perception of ABR risk and the expression of denial about this risk, r(282) = 0.454 *p* < 0.001. The more participants perceived ABR risk, the less they expressed denial about this risk. We also found a strong and positive relation between perception of ABR risk and concern for health consequences of this risk, r(286) = 0.521 *p* < 0.001: the more participants perceived ABR risk, the more they expressed concern for health consequences of this risk.

Regarding our third hypothesis, life science students expressed less denial about ABR risk (they had a higher score on denial scale) than mathematics students. We also found that life science students expressed more concern for health consequences of ABR risk than mathematics students did. Finally, life science students had a higher perception of ABR risk than mathematics students did (data summed up in Table 4).

We investigated the effect of the year of study on the expression of denial about ABR risk and found a principal effect of the year of study (first- to third-year) on the expression of denial about ABR risk, F(2, 283) = 14.18, *p* < 0.001, η^2^p = 0.091. Tukey’s post hoc tests revealed significant differences between participants in first-year (m = 24.99, SD = 4.62) and participants in second-year (m = 27.06, SD = 4.66), *p* = 0.010, 95% CI [−3.71, −0.42] and between participants in first-year (m = 24.99, SD = 4.62) and participants in third-year (m = 28.56, SD = 5.25), *p* < 0.001, 95% CI [−5.20, −1.94]. Denial was more prevalent among younger students than among older students (Figure 5). 

We also observed a principal effect of the year of study on the expression of concern for health consequences of ABR risk, F(2, 287) = 5.80, *p* = 0.003 η^2^p = 0.039. Tukey’s post hoc tests revealed significant differences between participants in first-year (m = 38.76, SD = 5.14) and participants in third-year (m = 41.10, SD = 4.93), *p* = 0.005, 95% CI [−4.09, −0.58]. Younger students were less concerned about health consequences than older students (Figure 5).

Finally, we observed a principal effect of the year of study on the perception of ABR risk, F(2, 303) = 3.80, *p* = 0.024 η^2^p = 0.024. Tukey’s post hoc tests revealed significant differences between participants in first-year (m = 46.41, SD = 9.59) and participants in third-year (m = 49.85, SD = 9.24), *p* = 0.025, 95% CI [−6.53, −0.35]. Younger students were less concerned about ABR risk than older students (Figure 6).

## 4. Discussion

The results were compared with those of a 2015 WHO survey of 9772 adults from 12 low (Egypt, India, Indonesia, Nigeria, Sudan, Vietnam) and high (Barbados, China, Mexico, Russian Federation, South Africa, Serbia) income countries [19]. The goal of our study was to assess the knowledge and applications of students at the University of Nîmes. First of all, the results of the questions on use and knowledge are very similar to former studies. We observed a low percentage of students that have taken antibiotics in the past month (14%). The WHO remarked that people in countries with higher income take fewer antibiotics than people living in countries with lower income. One possible explanation is that some antibiotics are available without a prescription in pharmacies [25,26]. Secondly, the same inaccuracies are observed about knowledge of ABR. They observed that 76% of participants still believe that the body rather than bacteria cause ABR. Moreover, 44% of respondents said that regular use of antibiotics was a prerequisite for becoming infected with a resistant strain, which is a second inaccuracy. For questions #4–6, we remarked that respondents have made more errors and misstatements concerning the prescription and use of antibiotics. We noticed that, in contrast to the previous study, where bladder infection was frequently chosen (75%) for question #7, only 14% chose gonorrhea, possibly due to a lack of knowledge about this disease. It is possible that the high rate of flu response is because secondary infections caused by the flu can be treated with antibiotics [27]. These findings corroborate Prigitano’s results. Younger students appear to be less aware of bacterial infection-related diseases than senior students.

We predicted that subjects who engaged in inappropriate behavior would be less concerned about the health consequences of ABR than those who engaged in appropriate behavior. We also assumed that they would be less concerned about the threat of ABR. This hypothesis appears to be corroborated. Indeed, we used three questions to determine which practices were appropriate and which were not. Only for question #6, we discovered that subjects who follow appropriate practices are more concerned about the health consequences of ABR risk and perceive ABR risk to be higher to a lesser extent. In addition, subjects who follow appropriate practices are less likely than those who follow incorrect ones to deny ABR risk (for questions #4 and #6). These findings appear to be consistent because people who use antibiotics appropriately may have more information, knowledge, and concern about ABR risk and its consequences. However, we noticed that 75% of participants answered correctly to questions #4 and #5 concerning self-medication. Furthermore, only 50% of participants answered correctly to question #6, which concerned suggesting to the doctor the possibility of repeating a previous prescription for the same symptoms.

We hypothesized that the more participants perceived ABR risk, the more concerned they would be about the health implications of this risk and the less denial they would express. This second hypothesis is also supported by the finding that there is a positive relationship between risk perception and health consequences, with higher perception of ABR risk corresponding to more concern for health consequences associated with this risk. We also discovered a link between the perception of ABR risk and the expression of denial of that risk. Indeed, the more participants are convinced that ABR is a high risk, the less denial they express.

These results suggest that people need to be aware of a risk to be aware of its health implications. Because ABR is a new risk, it has received less media attention. ABR is not perceived as a concrete and existing phenomenon, and this leads to a form of denial about the risk and its negative impacts (and reciprocally). This underlines the importance of communicating and informing the general public about this risk, especially as awareness of this risk can reduce self-medication. Finally, we expected to find differences between life science and mathematics students regarding the phenomenon of ABR. We hypothesized that life science students would have a higher perception of ABR risk and would be more concerned about the health consequences of this risk than mathematics students. We also stated that life science students would be less likely than mathematics students to deny ABR risk. Life science students were also found to be less likely to deny ABR risk than mathematics students.

When it came to risk perception, we noticed differences based on the level of expertise. Students following microbiology courses (as did the participants we considered as experts compared to mathematics students in our study) are more aware of the risk since their professors are more aware. This phenomenon has previously been observed and studied in another field: ecology. Differences between experts and naive subjects in knowledge utilization and mobilization to answer questions were shown [28].

We observe that the more specialist one is, the more one perceives the risk of this emerging danger. Information can only be considered as relevant when one is close to a reliable source of information, that is to say, a scientific source of information. Yet, not only do experts have access to this type of information but they are able to understand it. This is not always the case for naive people who require a popularized form of information since they have no pre-made knowledge of the domain. However, a trend was discovered in this research. Even experts are not immune to erroneous answers and practices [29]. Finally, we observed that third-year students expressed less denial of ABR risk than first-year students. We also noticed that third-year students were more concerned about the health consequences of ABR risk and perceived ABR risk to be higher than first-year students. These findings could indicate that students develop and improve their knowledge of ABR during and after their studies.

The limitations of the study are the sample size (Anovas performed in this study presented very low statistical power and must be considered as exploratory results), the age range and the educational level of respondents. The absence of a control group involving people who do not have a high level of education can be noted. Yet, our study underlined that there is an urgent need for information, education and awareness development about ABR risk.

## 5. Conclusions

Data obtained from 310 students of high level of education at the University of Nîmes, in urban areas, cannot be applied to the whole French population. Nevertheless, significant improvements have been made in order to enhance and shade the results of previous works. Nonetheless, no study on the French population has been conducted to date. The latter could be used as a complement to the WHO study to enrich our understanding of this emerging risk. Our findings suggest that it is critical to inform and communicate with the general public in a simple way about ABR risk. This requires a better awareness and knowledge about disease treatments and antibiotic therapies, regardless of the expertise field. This will help to prevent self-medication and promote appropriate practices. 

This study is part of a global awareness of this emerging risk that must be urgently considered. Indeed, the worrying evolution of this threat is dependent on appropriate practices and the awareness is crucial. It means dosage adherence and no self-medication behaviors are encouraged. 

## Figures and Tables

**Figure 1 ijerph-18-09692-f001:**
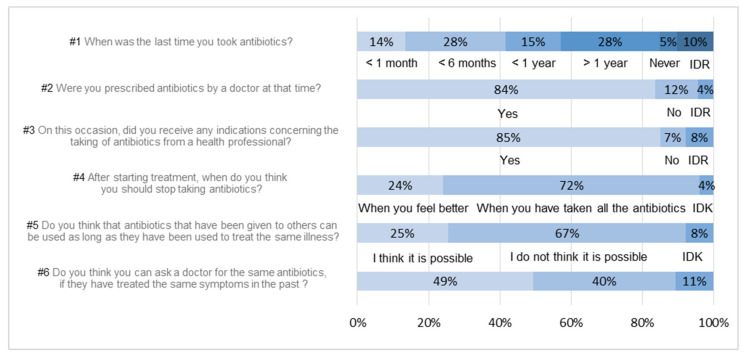
Percentages of response to first 6 questions.

**Figure 2 ijerph-18-09692-f002:**
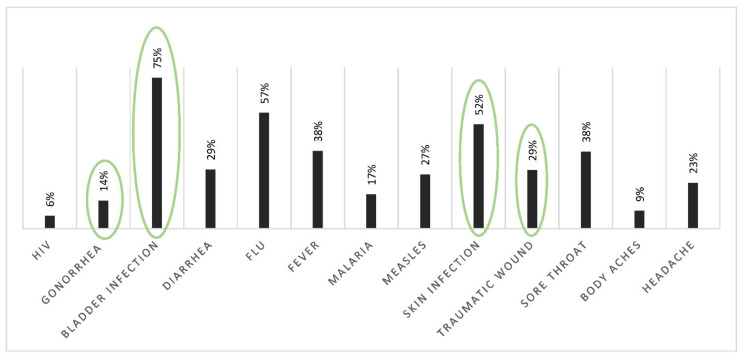
Percentage of responses to question #7: “Which of the following diseases/disorders do you think can be treated with antibiotics?”. Treatable with antibiotics when circled in green.

**Figure 3 ijerph-18-09692-f003:**
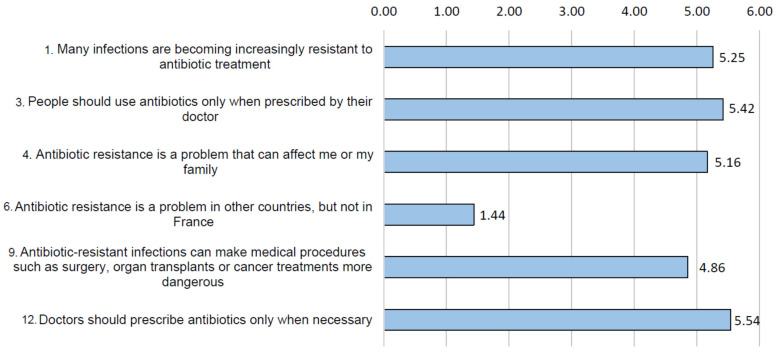
Mean score obtained by respondents for concern about risk perception and appropriate practices concerning ABR risk (question #8, items 1, 3, 4, 6, 9, 12; score from 1 to 6).

**Figure 4 ijerph-18-09692-f004:**
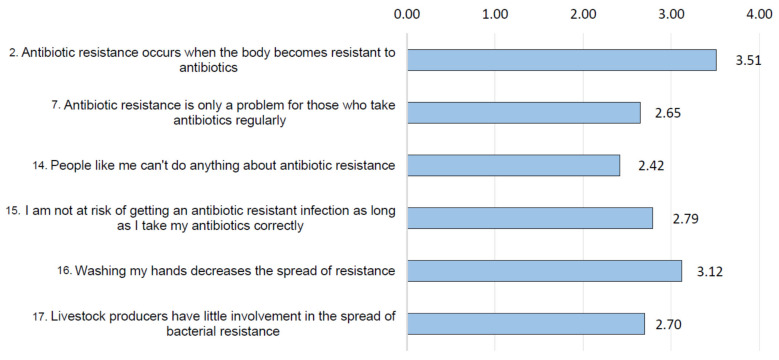
Mean score obtained by respondents about ABR knowledge and its transmission (question #8, items 2, 7, 14, 15, 16, 17; score from 1 to 6).

**Figure 5 ijerph-18-09692-f005:**
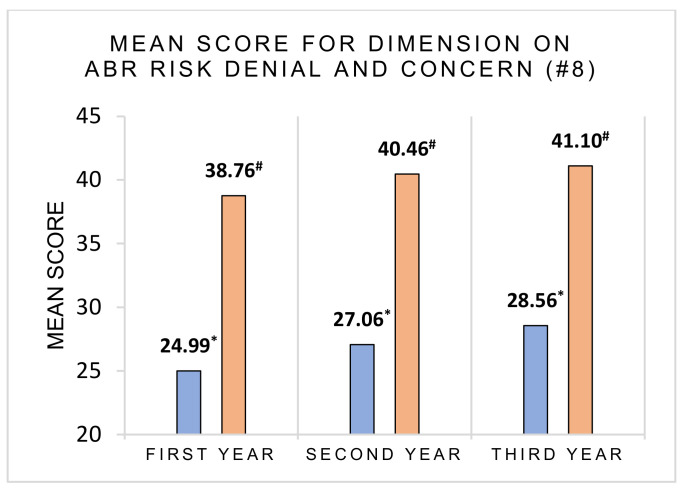
Expression of denial and health consequences for ABR risk depending on the year of study. The higher the score is, the lower the denial of the ABR risk and the higher the concern for health consequences of ABR risk. * Score is the sum of the answers to the items 1, 4, 5, 6, 9, 11, 12, 13 of question #8. ^#^ Score is the sum of the answers to the items 2, 7, 8, 14, 18, 19 of question #8.

**Figure 6 ijerph-18-09692-f006:**
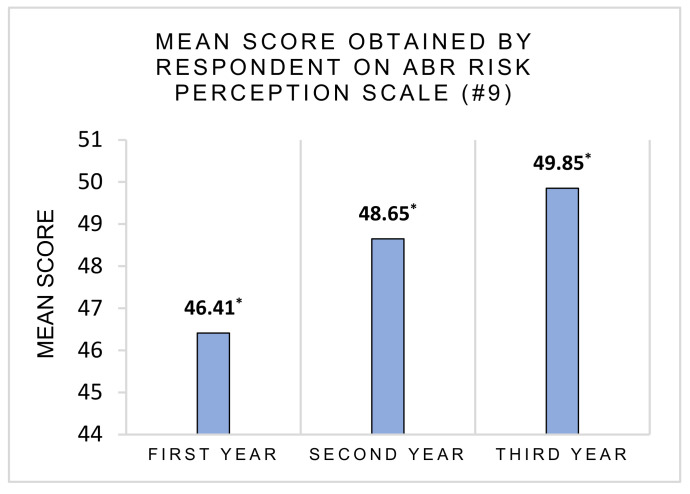
Perception of ABR risk depending on the year of study. The higher the score is, the higher the risk perception is. * Score is the sum of the answers to the items of question #9.

**Table 1 ijerph-18-09692-t001:** Pattern matrices of the concern for health consequences of ABR risk scale and the denial of ABR risk scale (EFA—Principal Component Analysis, Varimax rotation).

Item	Factor 1	Factor 2
1	**0.630**	0.046
2	−0.070	**0.521**
4	**0.588**	−0.070
5	**0.465**	0.172
6	**0.314**	0.275
7	0.053	**0.625**
8	0.385	**0.575**
9	**0.511**	0.220
11	**0.385**	0.277
12	**0.559**	−0.019
13	**0.688**	0.025
14	0.154	**0.555**
18	0.301	**0.399**
19	0.219	**0.437**

**Table 2 ijerph-18-09692-t002:** Pattern matrices of the risk perception of the ABR scale (EFA—Principal Component Analysis).

Item	Factor 1
1	0.749
2	0.785
3	0.570
4	0.336
5	0.386
7	0.481
8	0.600

**Table 3 ijerph-18-09692-t003:** First hypothesis data. D: Denial about ABR risk. HC: Health consequences of ABR risk. RP: Risk perception of ABR risk.

First Hypothesis: Subjects with Inappropriate Practices Would Have a Lower ABR Risk Perception, Concern and More Denial About It			
	Question #4	Question #5	Question #6
Good answer data	**D**: m = 26.90, SD = 5.17 **HC**: m = 40.24, SD = 5.03	**HC**: m = 40.32, SD = 5.15 **RP**: m = 48.61, SD = 9.89	**D**: m = 27.48, SD = 5.00 **HC**: m = 41.02, SD = 4.86 **RP**: m = 50.66, SD = 9.90
Wrong answer data	**D**: m = 24.94, SD = 4.33	**HC**: m = 38.88, SD = 5.27 **RP**: m = 46.12, SD = 8.84	**D**: m = 25.43, SD = 5.00 **HC**: m = 39.13, SD = 5.31 **RP**: m = 46. 24, SD = 8.72
Global data	**HC**: t(269) = −2.834, *p* = 0.005, d = 0.39, 95% CI [−3.31, −0.60]	**HC**: t(266) = −2.037, *p* = 0.043, d = 0.28, 95% CI [−2.82, −0.05] **RP**: t(281) = −1.939, *p* = 0.053, d = 0.26, 95% CI [−4.97, −0.04]	**D**: t(253) = −3.258, *p* = 0.001, d = 0.41, 95% CI [−3.29, −0.81] **HC**: t(259) = −2.962, *p* = 0.003, d = 0.37, 95% CI [−3.14, −0.63] **RP**: t(269) = −3.914, *p* < 0.001, d = 0.48, 95% CI [−6.66, −2.20]

**Table 4 ijerph-18-09692-t004:** Third hypothesis data.

Third Hypothesis: Students with a High Expertise in Life Science Would Have a Higher ABR Risk Perception, Concern and Lower ABR Risk Denial.			
	Denial about ABR Risk	Concern for Health Consequences	Perception of ABR Risk
Life science students	m = 27.04, SD = 5.06	m = 40.33, SD = 4.83	m = 48.87, SD = 9.39
Mathematics students	m = 23.81, SD = 3.86	m = 37.44, SD = 6.23	m = 44.11, SD = 9.09
Global data	t(284) = −4.566, *p* < 0.001, d = 0.67, 95% CI [−4.62, −1.84]	t(288) = −3.811, *p* < 0.001, d = 0.56, 95% CI [−4.39, −1.40]	t(304) = −3.673, *p* < 0.001, d = 0.51, 95% CI [−7.31, −2.21]

## Data Availability

All raw data can be found in Appendix A: Appendix A.

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
