# Peer review of "Evaluation of Knowledge and Risk Perception about Antibiotic Resistance in Biology and Mathematics Young Students in Nîmes University in France"

_ijerph, 2021, doi:10.3390/ijerph18189692_

Round 1
Reviewer 1 Report
The manuscript deals with an important topic concerning basic knowledge and risk perception about antibiotic resistance (ABR) among life science and mathematics students in Nîmes University. The study is well conducted and the cited literature is selected among 2000-2018. However I have some remarks related to the description of Materials and Methods section.
Special comments:
- The study population enrolled 310 participants. There were a small age range and all respondents represented higher education degree. Beside that, in the „Participants” section, I would advice to add numbers of students females i.e. 114 women (36,8%) AND 196 men (63,2%). It should be clarified for all life science and mathematics students by sex, number and percentage, mean age, standard deviation and age range. By the way, I would rather use words „men and women” in order to people. „Males and females” are used to describe sex in animals.
- Another concern regards to the participants’ enrollment to the study. How they were recruited? By the announcements on Faculty’s?
- I would advise that in Methods section should be mentioned that survey was divided on 3 parts and each of it was constructed of 7, 8 and 20 closed questions besides that full questionnaires have been added as Supplementary Materials.
- In the Conclusion section it would be worth to mention that as shown by the results of the survey, increasing the knowledge and awareness of disease treatment and antibiotic therapy for all social groups, regardless of their education, seems to be a good idea.
Author Response
Zohra Benfodda
University of Nîmes, Carmes
Place Gabriel Péri 30000 Nîmes
33 (0)4 66 27 95 89
Nîmes, 30 August 2021
Dear reviewers,
Thank you for your message regarding our manuscript named “Evaluation of Knowledge and Risk Perception about Antibiotic Resistance of Biology and Mathematics Young Students in Nîmes University in France, by Duvauchelle et al.
At first, we would like to thank you for your constructive comments and propositions to improve the manuscript.
In view of your comments, we are now submitting a revised version of the manuscript. You will find all the documents necessary for the revision: the revised draft with the corrections highlighted in yellow named draft “MDPI Duvauchelle revised”.
You will find below a table with the comments of the three reviewers and our answers.
We hope this revised version will be suitable for its final acceptance in International journal of environmental research and public Health.
We are now looking forward to hearing from you soon.
Please feel free to contact me if you have any queries.
Looking forward to hearing from you.
Sincerely yours,
Zohra Benfodda, Ph. D.
In this table, you will find the questions and/or remarks and our answers or propositions :
The manuscript deals with an important topic concerning basic knowledge and risk perception about antibiotic resistance (ABR) among life science and mathematics students in Nîmes University. The study is well conducted and the cited literature is selected among 2000-2018. However I have some remarks related to the description of Materials and Methods section. |
|
The study population enrolled 310 participants. There were a small age range and all respondents represented higher education degree. Beside that, in the „Participants” section, I would advice to add numbers of students females i.e. 114 women (36,8%) AND 196 men (63,2%). It should be clarified for all life science and mathematics students by sex, number and percentage, mean age, standard deviation and age range. By the way, I would rather use words „men and women” in order to people. „Males and females” are used to describe sex in animals. |
All suggested modifications were realized (Line 100 -104) Terms “Males” and “Women” were replaced by “Men” and “Women” |
Another concern regards to the participants’ enrollment to the study. How they were recruited? By the announcements on Faculty’s? |
Students were asked to answer a survey in the beginning of class in the University (≈10 minutes). No recruitment, all present students answered to the survey. |
I would advise that in Methods section should be mentioned that survey was divided on 3 parts and each of it was constructed of 7, 8 and 20 closed questions besides that full questionnaires have been added as Supplementary Materials. |
Modifications were added: Part one has been defined to correspond to question 1 to 7, second part to correspond to question 8 and third part to correspond to question 9. (Lines 114-115, Lines 116-117, Lines 126-127). |
In the Conclusion section it would be worth to mention that as shown by the results of the survey, increasing the knowledge and awareness of disease treatment and antibiotic therapy for all social groups, regardless of their education, seems to be a good idea. |
We agree with the reviewer. A sentence has been added as suggested (Line 379-381). |
Reviewer 2 Report
This original paper deals with a global issue that is on the WHO agenda of health topics deserving scientific attention and evidence-based interventions. The aims of this study are interesting.
Regarding the paper itself, in my opinion, some modifications are required:
- Figure 2 - should be added in the description of the figure: circled in green- treatable with antibiotics
- Figure 4 - the numbers in the figure should be sorted outThe authors do not inform about
- Ethical Considerations - whether the approval of the ethics committee has been obtained ?
Author Response
Zohra Benfodda
University of Nîmes, Carmes
Place Gabriel Péri 30000 Nîmes
33 (0)4 66 27 95 89
Nîmes, 30 August 2021
Dear reviewers,
Thank you for your message regarding our manuscript named “Evaluation of Knowledge and Risk Perception about Antibiotic Resistance of Biology and Mathematics Young Students in Nîmes University in France, by Duvauchelle et al.
At first, we would like to thank you for your constructive comments and propositions to improve the manuscript.
In view of your comments, we are now submitting a revised version of the manuscript. You will find all the documents necessary for the revision: the revised draft with the corrections highlighted in yellow named draft “MDPI Duvauchelle revised”.
You will find below a table with the comments of the three reviewers and our answers.
We hope this revised version will be suitable for its final acceptance in International journal of environmental research and public Health.
We are now looking forward to hearing from you soon.
Please feel free to contact me if you have any queries.
Looking forward to hearing from you.
Sincerely yours,
Zohra Benfodda, Ph. D.
In this table, you will find the questions and/or remarks and our answers or propositions :
Reviewer 2
|
|
This original paper deals with a global issue that is on the WHO agenda of health topics deserving scientific attention and evidence-based interventions. The aims of this study are interesting. |
|
Figure 2 - should be added in the description of the figure: circled in green- treatable with antibiotics |
This information has been added in the caption of Figure 2 (Lines 201-202). |
Figure 4 - the numbers in the figure should be sorted out |
Figure 4 has been revised. |
The authors do not inform about ethical Considerations - whether the approval of the ethics committee has been obtained ? |
Respondents were adults, the survey is anonymous and did not contain private information. Only oral approval from the respondent has been obtained but no approval from the ethics committee. |

Reviewer 3 Report
Antibiotic resistance is a phenomenon that demands studies from various angles. Firstly, it is important to know the population's knowledge about antibiotic resistance and, secondly, to inform in a simple way about the resistance risk. In addition, this work gives a new dimension - not only a description of general knowledge about antibiotic resistance, but also an assessment of students' social perceptions and behaviour (two dimensions). In the future, it would be appropriate to extend the sample to different age groups, areas and education levels.
I consider the article fit for publication. All these weaknesses are relatively minor and do not distract from the strength and importance of this study. I recommend:
- add numerical data comparing risk perception to the “Abstract”;
- add the used statistical analysis to the “Materials and Methods”;
- edit the “Results”: there is a lot of data on the three hypotheses in the text, it is confusing - it can be replaced by a table; figures 5 and 6 can be combined into one figure.
Author Response
Zohra Benfodda
University of Nîmes, Carmes
Place Gabriel Péri 30000 Nîmes
33 (0)4 66 27 95 89
Nîmes, 30 August 2021
Dear reviewers,
Thank you for your message regarding our manuscript named “Evaluation of Knowledge and Risk Perception about Antibiotic Resistance of Biology and Mathematics Young Students in Nîmes University in France, by Duvauchelle et al.
At first, we would like to thank you for your constructive comments and propositions to improve the manuscript.
In view of your comments, we are now submitting a revised version of the manuscript. You will find all the documents necessary for the revision: the revised draft with the corrections highlighted in yellow named draft “MDPI Duvauchelle revised”.
You will find below a table with the comments of the three reviewers and our answers.
We hope this revised version will be suitable for its final acceptance in International journal of environmental research and public Health.
We are now looking forward to hearing from you soon.
Please feel free to contact me if you have any queries.
Looking forward to hearing from you.
Sincerely yours,
Zohra Benfodda, Ph. D.
In this table, you will find the questions and/or remarks and our answers or propositions :
Reviewer 3 |
|
Antibiotic resistance is a phenomenon that demands studies from various angles. Firstly, it is important to know the population's knowledge about antibiotic resistance and, secondly, to inform in a simple way about the resistance risk. In addition, this work gives a new dimension - not only a description of general knowledge about antibiotic resistance, but also an assessment of students' social perceptions and behaviour (two dimensions). In the future, it would be appropriate to extend the sample to different age groups, areas and education levels. |
We agree with the reviewer remark and the idea to extend the study interest us. |
- add numerical data comparing risk perception to the “Abstract”; |
Numerical data were added in the abstract (Line 21-22 and 24). |
- add the used statistical analysis to the “Materials and Methods”; |
G*Power software has been used to realize the statistical analysis. A sentence has been added in Materials and Methods (Line 109). |
- edit the “Results”: there is a lot of data on the three hypotheses in the text, it is confusing - it can be replaced by a table; figures 5 and 6 can be combined into one figure. |
We agree with the remarks. Results part has been revised. Table 3 (Line 237) and Table 4 (Line 253) were added respectively to sum data information for first and third hypothesis. As second hypothesis part was short, no table was added. We hope modifications suit for a better understanding. Figure 5 and 6 were merged as suggested. |
